# SARS-CoV-2 and Dengue Virus Coinfection in a Mexican Pediatric Patient: A Case Report from Early Molecular Diagnosis

**DOI:** 10.3390/pathogens11111360

**Published:** 2022-11-16

**Authors:** Eloy A. Zepeda-Carrillo, Francisco M. Soria Saavedra, Maria B. Mercado Villegas, Alejandra López Alvarado, Jose Angel Regla-Nava, Rafael Torres-Valadez

**Affiliations:** 1Specialized Unit in Research, Development and Innovation in Genomic Medicine, Nayarit Center for Innovation and Technology Transfer, Autonomous University of Nayarit, Tepic 63000, Mexico; 2Tepic Specialty Hospital “Dr. Antonio González Guevara”, Health Services in Nayarit, Tepic 63000, Mexico; 3State Public Health Laboratory, Health Services in Nayarit, Tepic 63000, Mexico; 4Department of Microbiology and Pathology, University Center for Health Science (CUCS), University of Guadalajara, Guadalajara 44340, Mexico; 5Integral Health Academic Unit, Autonomous University of Nayarit, Tepic 63000, Mexico

**Keywords:** SARS-CoV-2, COVID-19, dengue virus, coinfection, molecular diagnosis, Mexico

## Abstract

Mexico is an endemic region for dengue virus (DENV). The increase in this disease coincides with outbreaks of COVID-19, both of which are single-stranded positive RNA viruses. These characteristics make it difficult to distinguish each disease because they share clinical and laboratory features, which can consequently result in misdiagnoses. This is why the use of precision confirmatory tests (qRT-PCR) are crucial for early diagnosis. We herein report a pediatric patient who presented a coinfection for DENV and COVID-19, “SARS-CoV-2/Dengue”. This patient initially presented a fever, cough, and headache and, three days later, developed generalized pain and epistaxis. Blood studies revealed thrombocytopenia and leukopenia, and the patient was admitted to the hospital for a probable DENV infection. Within 48 h, qRT-PCR tests specific for SARS-CoV-2 and DENV were performed and resulted as positive. The patient immediately received pharmacological treatment with azithromycin, oseltamivir, and metamizole. During hospitalization (9 days), the patient had no signs of respiratory distress and maintained normal body temperature and normal blood oxygen saturation. This case warns of the need for early diagnosis and adequate clinical and pharmacological management in the face of a “SARS-CoV-2/Dengue” coinfection. Early molecular detection of both viruses and timely treatment helped the patient to achieve a favorable recovery.

## 1. Introduction

Dengue virus (DENV) infection is an endemic disease in tropical and subtropical regions, including Mexico. This mosquito-borne virus causes >500,000 infections and 20,000 deaths annually [1]. DENV has four serotypes including dengue fever (DF), dengue hemorrhagic fever (DHF), and dengue shock syndrome (DSS), all which have been described [2]. DENV patients have nonspecific symptoms and show signs that are similar to infections caused by coronaviruses, including SARS-CoV-2, or arboviruses, including Zika and chikungunya. The endemic situation of DENV in Mexico coincided with the outbreak of the global COVID-19 pandemic [3]. Since the discovery of SARS-CoV-2, more than 7 million infections and 340,000 deaths have been reported in Mexico, one of the most affected countries in Latin America [4]. Some regions of several countries face high rates of DENV and COVID-19 infection, with an increased burden on health resources and systems.

Both infections have similar clinical manifestations and hematological laboratory results; therefore, the possibility of a coinfection generates a scenario of potential misdiagnosis and a lack of timely diagnosis [5,6,7,8]. Several studies have reported that DENV infection overlaps with COVID-19 infection and vice versa, particularly in the incubation phase [9,10,11]. Both viruses have a broad spectrum of clinical manifestations ranging from headache, fatigue, fever, myalgia, arthralgia, diarrhea, and skin rashes to more severe conditions including sepsis, hypovolemic shock, vasoplegia, cardiopulmonary collapse, and hyperactivation of the immune system [12,13,14]. The overlap in clinical and laboratory characteristics including leukopenia, lymphopenia, and thrombocytopenia mainly results in challenges of making correct diagnosis, managing, and pharmacology treatments according to both diseases, which are completely different [7,15]. Other studies have shown coinfections during the pandemic with other pathologies including hemoglobinopathies and hyperferritinaemia [16,17].

A fast, early, and accurate diagnosis for DENV and COVID-19 is convenient. Serological tests are useful and mainly used for a quick diagnosis and epidemiology studies. However, the accuracy of these tests is limited, and the possibility of cross-reaction is very high. The antigen-based test or antibody-based test for COVID-19 have recently emerged with their specificity and sensitivity not yet reported in large-scale studies. For antibody-based test, the limits of detection of IgM or IgG antibodies for SARS-CoV-2 variants depend on the course of the COVID-19 disease and integrity of the immune system of the patients [18]. This could lead to false-positive results and misdiagnosis, as well as delays in proper treatments of patients [19]. In this context, the use of confirmatory tests through PCR (polymerase chain reaction) to amplify the viral genetic component for both DENV and COVID-19 is crucial to obtain an early confirmatory diagnosis and adequate clinical attention [20]. Nevertheless, when possible, it is preferred to confirm the diagnosis by performing both qPCR and antibody tests.

However, information about DENV and COVID-19 coinfection is still limited. This has consequences for both patient care and public health, particularly in the accurate diagnosis of these cases. In this case report, we present and describe a female of 9 years of age who presented a nonspecific infection; molecular diagnostic tests were positive for both DENV and COVID-19, “SARS-CoV-2/Dengue”.

## 2. Case Presentation

A 9-year-old girl, without a prior history of dengue infection, developed a fever of 38.5 °C, dry cough, headache, and vomiting on 11 May 2020. After taking oral antibiotics (amoxicillin 60 mg/kg/day) and an analgesic (metamizole 10 mg/kg/day) for two days, her temperature persisted above 39 °C. She also developed odynophagia, myalgia, arthralgia, and epistaxis. On 13 May, a hematic biometry was performed and resulted in thrombocytopenia (11,000/μL) and leukopenia (3110/μL) (Table 1). The patient was then admitted to the Primary Health Care Center with a probable diagnosis of dengue virus infection. Upon admission, physical examination revealed mild hyaline rhinorrhea, dry cough, and the presence of diffuse erythematous punctiform eruption, predominantly in the limbs.

### 2.1. Diagnosis

On 14 May 2020, the State Public Health Laboratory confirmed a positive result for SARS-CoV-2 by qualitative real-time reverse transcriptase–polymerase chain reaction (qRT-PCR) using the patient’s oropharyngeal swab test. The next day, the same laboratory confirmed a positive result for the dengue virus serotype 2 and negative result for Zika and chikungunya viruses all by qRT-PCR assays (Figure 1). According to the diagnostic criteria, she was confirmed as a COVID-19 patient coinfected with dengue virus. During the clinical evaluation, the patient and family of the patient disclosed that they had no contact outside their household. However, the mother had shown symptoms including a dry cough and possible fever, which resulted in a positive SARS-CoV-2 oropharyngeal swab test result, without requiring hospitalization.

### 2.2. Treatment and Monitoring

On 16 May 2020, the 9-year-old patient was transferred to the Secondary Health Care Center, enabled for the management of patients with COVID-19. Upon admission, she presented normal vital signs, a temperature of 36.5 °C, oxygen saturation at 96%, did not receive supplemental oxygen support, and was medicated with oral metamizole. Lung auscultation reveled discrete bilateral baseline hypoventilation of posterior dominance, and the presence of petechiae was observed in the lower extremities. The patient received pharmacological treatment including ceftriaxone (100 mg/kg/day) and oseltamivir (60 mg orally every 12 h each). A new hematic biometry was performed, observing once again thrombocytopenia (17,000/μL) and neutrophilia (740/μL). Blood tests revealed AST 47 U/L, CK-MB 38.8 U/L, DHL 935 U/L, C-reactive protein (CRP) 3.0 mg/dL, and procalcitonin 0.6 ng/mL (Table 1). General urine examination was performed without finding alterations in its parameters. A high-resolution computed tomography (HRCT) of the chest was performed, and images revealed discrete basal infiltrates without convergence. Given the clinical history, molecular studies, and chest HRCT findings, the patient was admitted to an isolation unit as a COVID-19 case, and the antibiotic was changed to azithromycin (10 mg/kg/day), while oseltamivir continued as antiviral treatment.

During her hospitalization, the patient had no signs or symptoms of respiratory distress, maintained normal body temperature, and good blood oxygen saturation (above 92%). Blood tests revealed a discrete increase in platelets, neutrophils, and glycemia (Table 1). On 19 May, she was discharged without petechiae or epistaxis and was encouraged to quarantine at home for at least 14 days. The ambulatory management consisted in azithromycin and oseltamivir orally for 5 complete days. The SARS-CoV-2 RNA test by oropharyngeal swab was repeated in her follow-up visit on 28 May with a negative result.

## 3. Discussion

The Pacific Coast is where most cases of dengue infection occur, and is the place where the patient of this case resides. The low incidence of COVID-19 infections in these areas does not urge the population to adopt new COVID-19 containment measures, which consequently increases the risk of possible DENV-SARS-CoV-2 coinfections. In this case report, the patient was initially treated as a DENV infection; however, after the evolution of the symptoms and the implementation of a molecular diagnosis, the existence of a DENV-COVID-19 coinfection was determined, which we have colloquially called “SARS-CoV-2/Dengue”. Pathophysiological, DENV and COVID-19 share similar signs and symptoms, and both diseases can lead to hemophagocytic lymphocytosis, causing bleeding disorders and hypovolemic shock [21]. This relationship can cause serious health problems, delays in the early diagnosis of both viruses, and generation of false positives for DENV and COVID-19, leading to an increased infection rate in this type of population.

This is the first case of DENV-COVID-19 coinfection in a Mexican pediatric patient being reported in the scientific literature. The clinical evolution of this patient was similar to a previous report on a couple from Singapore, who presented persistent fever, cough, and thrombocytopenia. Both adult patients were initially positive for dengue (IgM and IgG), according to the serological tests, but when performing the molecular tests by RT-PCR, the results were negative for DENV and positive for COVID-19 [22]. The highlight of these findings is viral interference and the generation of false positives. In this case that we report, molecular tests for both viruses were carried out immediately, giving priority to COVID-19 testing. In this situation, viral interference was avoided and COVID-19 was detected early, allowing the patient to isolate herself and focus on her health. In addition, the spread of the virus among the community and among medical personnel was also prevented. There was immediate attention for specific diagnostic testing, health care, and implementing of pharmacological treatment. This, in addition to the absence of comorbidities such as obesity, asthma, or diabetes, helped the patient to achieve a favorable recovery and additionally avoided the development of complications or signs of respiratory distress.

The antimicrobial treatment applied to this patient was based on “Management of severe COVID-19: treatment of co-infections” with previous SARS-CoV experience in 2003, in which the World Health Organization (WHO) reports outbreaks of atypical pneumonia secondary to severe acute respiratory infection (SARI), and “Guideline on management of severe acute respiratory syndrome (SARS)”. They also report associations of viral and bacterial coinfections and increased incidence of sepsis and pneumonia. In their management flowchart, they suggest standard therapy for symptomatic patients, including coverage for atypical pneumonia with a macrolide such as azithromycin [23,24]. As countries advance vaccine applications against COVID-19, communication about the safety of vaccines is now more relevant than ever to avoid having SARS-CoV-2 severely impact dengue-endemic areas [25]. In this sense, cases such as the one described in this report can prevent the complication of a “SARS-CoV-2/Dengue” coinfection by having their COVID-19 vaccine, which has been shown to be safe and effective.

It is well-known that DHF and DSS are the leading causes of child death in some countries, including Mexico. DF causes mild symptoms such as fever, headache, rash, and nausea. In this case report, the humoral immune response produces antibodies that are protective against that specific serotype but does not neutralize other virus subtypes. A second infection with other serotypes commonly causes a more severe reaction, with symptoms such as persistent fever, thrombocytopenia, hemorrhagic disorders, and hypovolemic shock (DHF and/or DSS). This response is due to a hyper-reactive immune reaction that destabilizes all organ systems [26]. Various studies indicate that children can acquire cross-reactive antibodies through maternal passive immunity and thus develop complications from the first contact with DENV [27]. In this case, however, the patient and her mother did not report past dengue infections. In this context, the presence of DENV-COVID-19 coinfection could be promoting the possibility of severe symptoms, such as hematological disorders from the first exposure with DENV without the need of reinfection, a mechanism that is probably due to the viral interference phenomenon.

## 4. Conclusions

The potential increase in “SARS-CoV-2/Dengue” mainly in endemic areas could lead to a delay in the correct diagnosis and management of DENV and COVID-19. This can consequently result in an increased spread of these viruses, greater morbidity and mortality, further collapses of health systems, and possibly irreparable consequences. In regions with high dengue prevalence, the application of molecular diagnostic tests for both DENV and COVID-19 should be implemented in all patients with fever and other related symptoms for an accurate diagnosis. However, in places without easy access to qPCR tests, blood and swab samples should be stored properly to allow for proper SARS-CoV-2/Dengue diagnosis confirmation. This would allow proper clinical management to avoid severe dengue and increased COVID-19 cases, as well as to protect health personnel involved in the care of these patients.

## Figures and Tables

**Figure 1 pathogens-11-01360-f001:**
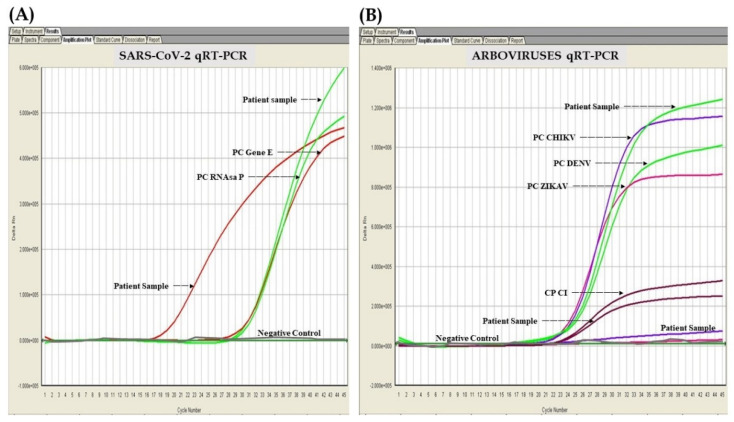
Amplification curves of SARS-CoV-2 and DENV by qRT-PCR. (**A**) Positive test by SARS-CoV-2 according to the Berlin Protocol (PAHO/WHO). (**B**) Positive test for dengue virus and negative test for Zika and chikungunya viruses using VIASURE MULTIPLEX Zika, Dengue & Chikungunya Real Time PCR Detection Kit (CerTest Biotec S.L. Spain). PC E Gene: positive control gene E; PC RdRp Gene: positive control RNA dependent transcriptase gene; PC CHIKV: positive control chikungunya virus; PC DENV: positive control dengue virus; PC ZIKAV: positive control Zika virus; PC IC: positive control internal control. Two independent experiments were performed.

**Table 1 pathogens-11-01360-t001:** Biochemical profile of the patient according to the clinical evolution of dengue virus and COVID-19 coinfection.

	13 May	16 May	17 May	18 May	Reference Value
WBC (10^3^/μL)	3.11	4.82	6.67	5.04	3.9 to 10.0
Neutrophils 10^3^/μL	47.3	0.74	1.39	1.28	1.56 to 6.13
Lymphocytes (10^3^/μL)	(ND)	(15.4)	(21.0)	25.4	34.0 to 71.1
Monocytes (10^3^/μL)	40.5	3.40	4.47	3.12	1.18 to 3.74
Platelets (10^3^/μL)	11.6	0.52	0.57	0.41	0.24 to 0.86
RBC (10^6^/μL)	11	17	50	126	180 to 300
Hb (g/dL)	4.44	4.51	4.57	4.96	3.93 to 5.22
HTO (%)	12.3	12.1	12.4	13.2	11.2 to 15.7
Glucose (mg/dL)	37.1	36.6	36.7	39.7	34.1 to 44.9
Urea (mg/dL)	ND	95	109	111	76 to 106
Creatinine (mg/dL)	ND	31.4	20.9	25.9	14.9 to 38.5
AST (U/L)	ND	0.46	0.39	0.39	0.6 to 1.0
ALT (U/L)	ND	47	38	49	<32
Albumin (g/dL)	ND	20	20	33	<33
ALP (U/L)	ND	4.1	4.0	4.3	3.5 to 5.2
CK-MB (U/L)	ND	151	132	134	35 to 105
LDH (U/L)	ND	38.8	ND	ND	7 to 25
PCR (mg/dL)	ND	935	ND	ND	240 to 480
PCT (ng/mL)	ND	3.0	ND	ND	<5.0
WBC (10^3^/μL)	ND	0.6	ND	ND	Low risk < 0.5 High risk > 2.0

WBC: white blood cell; RBC: red blood cell; Hb: hemoglobin; HTO: hematocrit; AST: aspartate aminotransferase; ALT: alanine aminotransferase; ALP: alkaline phosphatase; CK-MB: creatine kinase-MB fraction; LDH: lactic dehydrogenase; PCR: protein “C” reactive; PCT: procalcitonin; ND: no data.

## Data Availability

The data are not publicly available due to privacy.

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
