# Peer review of "SARS-CoV-2 and Dengue Virus Coinfection in a Mexican Pediatric Patient: A Case Report from Early Molecular Diagnosis"

_pathogens, 2022, doi:10.3390/pathogens11111360_

Round 1

Reviewer 1 Report

Here are some recommendations:

1. Remove the term "coronadengue", I consider SARS-CoV-2/ Dengue coinfection is more formal

2. line 37 change ´fever dengue´ to dengue fever

3. line 40 it seems your are including coronavirus as an arbovirus, correct the grammar

4. line 165 revise grammar

5. You gave the patient antibiotics even when the viral infection was confirmed, explain the reason

6. How many times did you performed the qPCR to be sure is not a false positive?, which is very common. Where are the negative controls in the figure?

7. Include the serological test to confirm SARS-CoV-2 and DENV if you have it. If you have the patient serum stored, perform both tests to have more certainty of the co infection.

Author Response

Reviewers' comments:

Reviewer #1 (Remarks to the Author):

  1. Remove the term "coronadengue", I consider SARS-CoV-2/ Dengue coinfection is more formal.

We thank the reviewer for this suggestion. Following the reviewer’s recommendations, we edited the title and now it is: “SARS-CoV-2 and Dengue Virus Coinfection in a Mexican Pediatric Patient. A Case Report from Early Molecular Diagnosis” (page 1, lines 2-3).

  1. line 37 change ´fever dengue´ to dengue fever.

We thank the reviewer for this comment and following this suggestion, we have changed it (page 1, line 37).

  1. line 40 it seems your are including coronavirus as an arbovirus, correct the grammar.

Following the reviewer’s suggestion, we have edited the sentence with appropriate clarification: (page 1, lines 39-41). DENV patients have non-specific symptoms and show signs that similar to infections caused by coronavirus including SARS-CoV-2 or arboviruses including Zika and Chikungunya.

  1. line 165 revise grammar.

We thank the reviewer for this suggestion. This change is reflected in the new version of the manuscript (page 5, line 240-242). In this situation, viral interference was avoided and COVID-19 was detected early, allowing the patient to isolate herself and focus on her health. In addition, the spread of the virus among the community and among medical personnel was also prevented.

  1. You gave the patient antibiotics even when the viral infection was confirmed, explain the reason.

Following the reviewer’s comment, this is reflected in the discussion (page 5, line 248-252) as discussed in the new manuscript. The antimicrobial treatment applied to this patient was based on “Management of severe COVID-19: treatment of co-infections”1with previous SARS-CoV experience in 2003, WHO reports outbreaks of atypical pneumonia secondary to SARI, and “Guideline on management of severe acute respiratory syndrome (SARS)2. In the next sentences is reflected (page 5, line 253-255).  In their management flowchart, they suggest standard therapy for symptomatic patients including coverage for atypical pneumonia with a macrolide such as azithromycin.  

  1. How many times did you performed the qPCR to be sure is not a false positive?, which is very common. Where are the negative controls in the figure?

We appreciate this comment. We performed two independent experiments from qPCR with a same result (positive), negative controls were used. This information is reflected in the new version of the figure 1.

  1. Include the serological test to confirm SARS-CoV-2 and DENV if you have it. If you have the patient serum stored, perform both tests to have more certainty of the co infection.

We appreciate this general observation. Unfortunately, we do not have serum stored from this patient to include the serological test.

Reviewer 2 Report

This case report is well written, the first report of pediatric patients with Dengue Virus and COVID-19 co-infections and adds to current knowledge, with emphasis on the challenges and necessity of precise diagnosis of co-infection, to circumvent the potentially fatal outcomes and socioeconomic impact on humankind.  

I have a few minor comments and suggestions for authors to consider

The title accurately described the case however,  full-stop after [ “Coronadengue”.  ] seems like a typographical error. Also the authors should consider including the phrase “a case report” in the title as per the CARE guidelines for case reports.

The Introduction needs some work. Line 45-47: I suggest expanding/citing more relevant, several case reports of DENV-Covid 19 coinfection which have been published in the last three years, with their clinical and laboratory characteristics, limitations on diagnosis and treatments.

Line 59-60: I suggest authors should provide additional information about the general serological diagnostic tests specific to the both infection diagnosis in question.Further elaborate on the challenges of misdiagnosis via serological tests and in turn importance of defining advanced rapid and accurate diagnostics.

I suggest to well structure the Case Presentation section, incorporating/dividing it into three subsections - Case presentation , Investigation, and diagnosis/treatment

For Table 1 , %Neutrophils 103/uL can be rewritten as Neutrophils 103/uL (%)

Overall, Discussion and conclusion are well written.

Author Response

Reviewer #2 (Remarks to the Author):

This case report is well written, the first report of pediatric patients with Dengue Virus and COVID-19 co-infections and adds to current knowledge, with emphasis on the challenges and necessity of precise diagnosis of co-infection, to circumvent the potentially fatal outcomes and socioeconomic impact on humankind. 

We thank the reviewer for this comment.

I have a few minor comments and suggestions for authors to consider.

  1. The title accurately described the case however, full-stop after [ “Coronadengue”] seems like a typographical error. Also the authors should consider including the phrase “a case report” in the title as per the CARE guidelines for case reports.

We thank the reviewer for this comment and following this suggestion, we edited the title and now it is: “SARS-CoV-2 and Dengue Virus Coinfection in a Mexican Pediatric Patient. A Case Report from Early Molecular Diagnosis” (page 1, lines 2-3).

  1. The Introduction needs some work. Line 45-47: I suggest expanding/citing more relevant, several case reports of DENV-Covid 19 coinfection which have been published in the last three years, with their clinical and laboratory characteristics, limitations on diagnosis and treatments.

We would like to thank the reviewer for the highly positive and helpful comments to improve the manuscript. We expanded the citing (3-9). These changes are reflected in the new version of the manuscript. (page 2, line 62-74).

  1. Line 59-60: I suggest authors should provide additional information about the general serological diagnostic tests specific to the both infection diagnosis in question. Further elaborate on the challenges of misdiagnosis via serological tests and in turn importance of defining advanced rapid and accurate diagnostics.

We thank the reviewer for this suggestion. In the introduction, we mention the need to seek an accurate and early diagnosis of SARS-Cov-2 and/or Dengue virus infection through molecular methods and not blindly trust the result provided by a serological test considering all the limitations they present, (page 2, lines 77-78) and its importance is highlighted in the discussion (page 5, lines 202-205). Further, the antigen-based test or antibody -based test by COVID-19 have recently emerged with their specificity and sensitivity not yet reported in large-scale studies. For anti-body-based test, the limits of detection of IgM or IgG antibodies for SARS-CoV-2 variants depend on the course of the COVID-19 disease and integrity of immune system of the patients10. This could lead to false-positive results and misdiagnosis, as well as delays in proper treatments of patients11. In this context, the use of confirmatory tests through PCR (Polymerase Chain Reaction) to amplify the viral genetic component for both DENV and COVID-19 is crucial to obtain an early confirmatory diagnosis and adequate clinical attention12. However, information about DENV and COVID-19 coinfection is still limited. This has consequences for both patient care and public health, particularly in the accurate diagnosis of these cases. In this case report, we present and describe a female of 9 years of age who presented a nonspecific infection; molecular diagnostic tests were positive for both DENV and COVID-19, “SARS-CoV-2/Dengue”. This change is reflected in the new version of the manuscript (page 2, lines 79-92).

  1. I suggest to well structure the Case Presentation section, incorporating/dividing it into three subsections - Case presentation, Investigation, and diagnosis/treatment.

Following the reviewer’s suggestion, point 2 now includes the following subsections: case presentation page 2, line 94), diagnosis (page 2, line 105) and treatment and monitoring (page 3, line 171).

  1. For Table 1, %Neutrophils 103/uL can be rewritten as Neutrophils 103/uL (%).

Following the reviewer’s suggestion, we modified it in the Table 1.

Overall, Discussion and conclusion are well written.

We thank the reviewer for this comment.

Round 2

Reviewer 1 Report

Thanks for the corrections. Just one final suggestion, It would be better if you mention that both tests should be performed, qPCR and antibodies, maybe not in a daily basis diagnosis in the future in the case of an outbreak but as a recommendation for the characterization of the coinfection; I am aware that in some places where arboviruses are endemic is not posible to run qPCR easily but samples (blood and swabs) should be store properly to confirm later coinfections.

Author Response

Reviewers' comments:

Reviewer #1 (Remarks to the Author):

  1. Thanks for the corrections. Just one final suggestion, It would be better if you mention that both tests should be performed, qPCR and antibodies, maybe not in a daily basis diagnosis in the future in the case of an outbreak but as a recommendation for the characterization of the coinfection; I am aware that in some places where arboviruses are endemic is not possible to run qPCR easily but samples (blood and swabs) should be store properly to confirm later coinfections.

We thank the reviewer for this suggestion. This change is reflected in the new version of the manuscript. Nevertheless, when possible, it is preferred to confirm the diagnosis by performing both qPCR and antibody tests (page 2, line 88 and 89). However, in places without easy access to qPCR tests, blood and swab samples should be stored properly to allow for proper SARS-CoV-2/Dengue diagnosis confirmation
